# Dilemmas for Natural Living Concepts of Zoo Animal Welfare

**DOI:** 10.3390/ani9060318

**Published:** 2019-06-05

**Authors:** Mark James Learmonth

**Affiliations:** Animal Welfare Science Centre, The University of Melbourne, Parkville 3010, VIC, Australia; mlearmonth@student.unimelb.edu.au

**Keywords:** animal ethics, natural living, wilding, zoo animals, human-animal interactions, animal welfare

## Abstract

**Simple Summary:**

This ethical discourse specifically deals with dilemmas encountered within zoological institutions, namely for the concept of *natural living*, and a new term—*wilding*. *Wilding* refers to extrapolation of the natural living concept to *treating an animal as wild, residing in a wild habitat*. The problems associated with *wilding* are detailed. Complexities of natural living versus natural aesthetics as judged by humans, as well as the possibility of innate preference for naturalness within animals are examined. It is argued that unintended and unavoidable genetic and epigenetic drift favouring adaptations for life in a captive environment may still occur, despite zoos best efforts to prevent this from occurring. This article further discusses the blurred lines between *natural* and *unnatural* behaviours, and the overlaps with more important *highly-motivated behaviours*, which may be better predictors of positive affective states in captive animals, and thus, better predictors of positive well-being and welfare. Finally, as we are now in the *Anthropocene* era, it is suggested that human-animal interactions could actually be considered natural in a way, and notwithstanding, be very important to animals that initiate these interactions, especially for “a life worth living”.

**Abstract:**

This ethical discourse specifically deals with dilemmas encountered within zoological institutions, namely for the concept of *natural living*, and a new term—*wilding*. It is agreed by some that zoos are not ethically wrong *in principle*, but there are currently some contradictions and ethical concerns for zoos *in practice*. Natural living is a complicated concept, facing multiple criticisms. Not all natural behaviours, nor natural environments, are to the benefit of animals in a captive setting, and practical application of the natural living concept has flaws. Expression of natural behaviours does not necessarily indicate positive well-being of an animal. Herein it is suggested that *highly-motivated behaviours* may be a better term to properly explain behaviours of more significance to captive animals. *Wilding* refers to extrapolation of the natural living concept to *treating an animal as wild, residing in a wild habitat*. This definition is intrinsically problematic, as quite literally by definition, captivity is *not* a wild nor natural environment. Treating a captive animal *exactly the same as a wild counterpart* is practically impossible for many species in a few ways. This article discusses complexities of natural living versus natural aesthetics as judged by humans, as well as the possibility of innate preference for naturalness within animals. Zoos nobly strive to keep wild animals as natural and undomesticated as possible. Here it is argued that unintended and unavoidable genetic and epigenetic drift favouring adaptations for life in a captive environment may still occur, despite our best efforts to prevent this from occurring. This article further discusses the blurred lines between *natural* and *unnatural* behaviours, and the overlaps with more important *highly-motivated behaviours*, which may be better predictors of positive affective states in captive animals, and thus, better predictors of positive well-being and welfare. Finally, as we are now in the *Anthropocene* era, it is suggested that human-animal interactions could actually be considered natural in a way, and notwithstanding, be very important to animals that initiate these interactions, especially for “a life worth living”.

## 1. Introduction

To preface this article, I would acknowledge and address the implicit assumptions about animal welfare science and philosophy that have brought us to the ethical position herein. I would refer the readers to other published articles which explore the history of animal welfare and ethics in much depth, as these are used as a basis for our understanding and arguments [1,2,3,4,5,6,7,8,9,10]. This article specifically deals with competing ideals of optimal animal welfare within zoological institutions, namely concepts of *natural living*, and a new term—*wilding*. This discourse does not necessarily apply to other captive animal industries such as farms or laboratories.

As other ethicists have written, I agree that zoos are not ethically wrong *in principle*, but there are currently some contradictions and ethical concerns for zoos *in practice* [5]. It should be understood that I am a supporter of zoological institutions and their betterment, although I may disagree with some zoo practices, and between myself and other researchers in ethical views of the specific dilemmas herein. It is also acknowledged that zoos are not going away any time soon (see [5]), so it is the pragmatic duty of researchers and philosophers to work *with* zoos constructively. I would also like to acknowledge the positions of other researchers in the field, such as Weary and Robbins [9], and Yeates [10]. This article is not intended as a refutation of these other recent articles about natural living and holistic welfare, but rather to present an alternate conception of one part of overall animal welfare that may have been misconstrued in certain zoo environments, leading to in practice incongruence and dilemmas. I acknowledge that my arguments are formulated from my moral and ethical position that humans have an *obligation for special protection of captive animals*, especially zoo animals, and I subscribe to many (but not all) elements of *compassionate conservation* ethical theory of contemporary philosophers such as Bekoff [11] and Gray [12] over purely utilitarian or consequentialist approaches. At the moment *compassionate conservation* remains very anti-zoo in its position, however, as Gray [12] posits, there is much merit in using this ethic to work *with* zoos constructively, to enhance zoos’ ethics and practices.

I acknowledge the currently accepted academic focus on the three conceptual frameworks (orientations) of animal welfare: biological functioning, affective states, and natural living [2,13]. This is how the *science* of animal welfare is commonly taught to undergraduate and postgraduate learners in our discipline. It is acknowledged, however, that this not the only way to conceptualise the entire picture of captive animal welfare [3,9], and that these three conceptual frameworks do not encompass all relevant information in all situations. I acknowledge that a predominant model for characterising and assessing good welfare, especially within zoos, is the Five Domains Model of Mellor and Reid [14] and Mellor and Beausoleil [15]. Whilst incorporating pluralistic scientific elements of welfare, at its core the Five Domains Model assumes a hedonistic priority of animal welfare, that is, what the animal feels about its life and environment is the most important factor in holistic welfare. In this article a pluralistic basis of welfare is acknowledged, though for the sake of argument a hedonistic basis is prioritised. It is understood, however, that a hedonistic priority also misses some of the whole picture [3]; hedonism-based welfare conceptions are not dogma. The two scientific concepts of biological functioning and affective states will only be touched on in this article, as my primary focus is to shine a light on how the concept of *natural living* may have been pushed past its useful bounds in zoo situations.

It should be stated that whilst hedonistic conceptions of welfare are mostly concerned with “how the lives of sentient animals are going, for the sake of, and from the perspective of, the animals themselves” [6], it is strongly suggested here that (as written by Weary and Robbins [9]) *relationships matter*. That is, not only are the self-derived internal states of the individual highly important, but also those emotion-inducing relationships that are important to the individual—such as relationships to conspecifics, other animals, and humans including carers and visitors—*and* some relationships that *others* have *with* that individual may also be important to welfare outcomes (for example, the specific values and attitudes a person holds will affect their relationship with an individual animal, and reinforcers to this relationship create a bi-directional, perpetual feedback loop). These relationships may then be *reflected* by the internal affective states of both (or all) agents in that interaction [16,17]. This has been characterised by the general Hemsworth-Coleman model of human-animal interactions [18,19]. The general model has been specifically adapted for zoo visitor-animal relationships [20], pictured below (Figure 1). A very similar model has been proposed for zookeeper-animal interactions as well [20]. Human values and attitudes towards animals, and the relationships formed between them, can strongly influence subjective (hedonistic) experiences of welfare.

## 2. Natural Living

Natural living is a (sometimes) useful key concept in the assessment of animal welfare, often defined as “providing opportunities for animals to engage in natural, species-specific behaviours” [1,2,10]. As a concept, it suggests that animals’ well-being may be considerably improved if they are *able to perform species-specific behaviours from their natural repertoire*, especially innate behaviours. In practice, this has often been achieved by removing restrictions to these behaviours (whether they are physical or environmental restrictions) and by providing appropriate objects, resources or enclosures with/in which to perform the behaviour(s). Often, definitions of the concept also include phrases about housing animals in *natural environments*. However, not all natural behaviours, nor natural environments, are to the benefit of animals in a captive setting, and practical application of the concept has many flaws. A main criticism of the use of natural living has been that “the concept of natural is usually too poorly defined to provide a sound basis for animal welfare assessment, and thus when applied uncritically it may lead to poorer welfare instead of an improvement” [21]. This criticism has been expressed quite commonly in the past few decades [1,2,10,19,22].

Articulation of the concept, and its transposition to practical application in many captive settings have somewhat missed the point entirely. *Natural behaviour*, *natural living* and *naturalness* are poorly-defined key terms that are too often conflated with other concepts and measures of an animal’s overall well-being, such as feelings (affective state) or function (biological functioning) [10]. Expression of natural behaviours does not necessarily indicate positive well-being of an animal; likewise absence of some natural behaviours does not necessarily indicate suffering [23,24]. Nor should the term *natural behaviour* be used when actually referring to other conceptual types of behaviours, such as *highly-motivated behaviours*, which may be natural or unnatural, however there is often significant overlap between these two terms. Herein I will suggest that *highly-motivated behaviours* may be a better term to properly explain behaviours of more significance to captive animals, and discuss where boundaries between *harmless* and *harmful* highly-motivated behaviours may lie (as we still have an ethical obligation to protect animals from harming themselves, whether intentionally or accidentally, in captivity).

## 3. Wilding: The Natural Living Dilemma

*Natural living* has been a useful tool for improving welfare, but its practical application, especially within zoos, has been extended beyond its theoretical usefulness, and in many instances has been misinterpreted as what I will herein refer to as *wilding*. Wilding is a new term created to refer to extrapolation of the natural living concept to *treating an animal as wild, residing in a wild habitat*. Wild here refers to “living or growing in the *natural* environment; not domesticated or cultivated” [25]. From a decade of first-hand experience within the zoo industry, this *wilding* conception of natural living has been encountered often enough to be considered pervasive amongst many zoo personnel’s implicit beliefs and taught knowledge about how zoos should approach animal welfare, though actual prevalence rates have not been systematically investigated. Indeed, many welfare assessment and monitoring tools deployed by zoos focus somewhat on natural environments and natural behaviours [26]. This *wilding* conception is intrinsically problematic for any captive animal industry (especially zoos) as, quite literally by definition, captivity is *not* a wild nor natural environment [27]. To place a wild animal in an artificial environment (no matter how accurate a recreation of a natural setting) and still presume to treat it *exactly the same as a wild counterpart* is practically impossible for many animal species, in a few obvious ways.

Firstly, truly *wild* animals in nature are not *treated* by humans in a particular way—they are not under the direct care of humans, however, they may yet be influenced by humans [28]. These wild animals may be exposed to humans in multiple situations, and even have interactions with humans, but their lives are not solely dictated by humans as captors/guardians. This does not preclude the possibility of interactions (both positive and negative) or conflicts arising between humans and animals, animals venturing into “human spaces”, or encroachment of humans into an animal’s native space [28]. However, as soon as an animal is placed in captivity, no matter how wild its behaviours or instincts, its care (and indeed its survival) is then determined and controlled by those humans that placed it there. A person cannot place an animal in a captive environment then refrain from providing basic cares or resources (such as food, water and shelter), and yet expect the animal to survive, let alone to thrive. Even in a highly accurate recreation of a natural environment, those basic resources must still be provided by the controlling humans—that is, the environment has been created and curated to provide those resources for the animal, through natural or artificial structures.

Secondly, even if it were the case that humans could provide a perfect replica of an animal’s wild environment with wild conditions, would it be morally or ethically permissible? Would it be (morally) right? Forgetting for a second that this perfect replica would still have been constructed upon another natural or wild environment (thereby destroying a natural habitat and causing displacement of many native species), if truly a replica of natural conditions, then the animals placed in this environment would be subject to both the boons and significant hardships of nature. Nature is often bountiful and has allowed the rise of an amazingly diverse array of living beings, but has also borne witness to countless extinctions and ecological changes. Wild animals often must endure very harsh conditions to survive—conditions that objectively lead to periods of very poor welfare, when measured through scientific welfare concepts (biological functioning and affective states) [10,28,29,30]. Inclement weather and natural disasters such as droughts, fires or floods, are all common occurrences in nature. Animals must endure a lack of shelter, food or water in many areas; they must avoid predation, injury, and disease; they may experience miscarriages, offspring mortalities or reproductive issues; they often have to compete with other animals (both of their own and other species) for access to resources; and they have to navigate oft-unfair social interactions and hierarchies. Often, living in nature leads to *prolonged suffering* and ends in *premature death* for individuals.

Many *wild-type* or *natural behaviours* are also maladaptive in a captive environment (such as fratricide or infanticide for extreme examples; to significant inbreeding in closed populations; group ostracism of certain individuals; or unfulfillable migratory behaviours/motivations) [12,30,31]. Thus, if it was indeed the objective of captive animal industries, such as zoos, to perfectly replicate natural environments so their animals may live *wildly*, it follows that all of the hardships of nature would also occur, or would have to be imposed. This is not a tenable ethical position that any zoo organisation is known to advocate. Instead, natural recreations of wild environments in zoos try to focus mainly on *positive* elements of nature, without imposition of events or states that may significantly diminish the animal’s well-being [5,26,30,32]. Ethically, one will not find much (or any) opposition to this mode of treatment of the captive animals. This also provides a pro-captivity argument against some anti-captivity, *animal freedom*-based philosophies—captivity does indeed curtail some freedom of the captive animal, but it also provides solace and shelter from significant welfare-affecting hardships, which may be especially of benefit to those animals whom are most vulnerable to suffering. Indeed, if captivity is providing all of the needs and wants of an animal (including positive affective experiences), but without liberty, then liberty is not necessarily a basic interest of the animal [5]. Zoos are often the last bastion of hope for many endangered species, as their wild homes have been irreparably damaged or overtaken by ever-expanding human populations [12,28,30,33]. This is an ever more salient point after the United Nations Intergovernmental Science-Policy Platform on Biodiversity and Ecosystem Services (IPBES) released a 2019 report which estimates that anthropogenic influences may cause the extinction of 1 million species of animals and plants [33].

It should be noted that whilst zoos tend to focus mostly on recreating positive elements of nature and reducing *negative* circumstances, many zoos also understand the impossibility of complete elimination of all negative circumstances, events, or negative feelings within an animal. In fact, many zoos will impose slight negative circumstances if it is believed that they may be of benefit to the animals’ health, fitness, or experience of life [34]. That is, *harmless* or *minimally harmful* negative circumstances are sometimes imposed to increase stress resilience and/or physiological arousal of an animal [35,36]. For example, it has been reported that reliably signalling startling husbandry events can improve stress resilience and welfare of zoo-housed capuchins (*Sapajus apella*), whilst still leading to physiological arousal within the animals [37]. However, where is the distinction drawn between harmless, minimally harmful and very harmful negatives? And who makes these categorical judgements?

Through collaborative practices shared between many zoos, a few common circumstances for imposing *minimally harmful* negative events include: rotational predator-prey housing (where predatory species and prey species are rotated into the same enclosure at separate times); predator-prey adjacent housing with visual proximity; olfactory proximity between predator-prey species or dominant-subordinate species (sometimes in the form of “enrichment”, like adding predator bedding material to a prey enclosure); or auditory proximity between predator-prey species (such as housing prey species within earshot of vocalising predators or dominant species, or playing recorded audio of predator/dominant animal vocalisations near prey/subordinate species) [26,31]. These circumstances are thought to confer some resilience to animals through arousal of certain fear and vigilance responses, which can have a wide range of beneficial physiological effects, if not experienced for prolonged periods (acute stressors versus chronic stressors) [35,36]. Therefore, some mild harms are actually of high instrumental value within a captive environment. However, there should be a trepidation of pushing such stress responses too far in prey species, or causing inadvertent frustrations to these animals, for example in adjacent predator-prey housing where predators can visually see prey in very close proximity, but not actually reach them. Repeated frustration of consummatory outcomes may lead to development of negative affective states, as indicated by frustration-type behaviours [24,34]. More evidence is needed of the overall effects of the imposition of these stressors on individual animals, to ensure that the intended arousal and stress resilience is being achieved whilst avoiding unintended frustrations or development of negative affective states in these animals.

## 4. Natural Living, or Just Natural Looking?

Erstwhile, when considering and implementing positive natural enclosures, zoos may tend to focus only on those that, aesthetically, lead people to *believe* that the environment is *natural*. For example, lush plant-life (or well-designed arid/desert habitats), water features, painted backdrops or “mock-rock” walls, absence of artificial structures, and/or limiting contact with visitors (or even staff/keepers) whether the limitations are visual, tactile or proximal. Much of the time, considerations of what is aesthetically pleasing may eclipse considerations of what is functional and appropriate, with respect to evidence-based practices [31]. More than just *looking* natural, zoo animals’ enclosures must be able to provide necessary features and structures to allow animals to display a range of important behaviours, provide access to perform positive husbandry practices, and allow ease-of-access for emergency procedures to be adhered to (for both animal emergencies, and other visitor or human emergency situations which may occur). If a natural look is considered forefront, this may lead to functional inadequacies in many enclosures. Sometimes artificial structures in enclosures may be more appropriate to facilitate specific animal behaviours—whereas natural structures may weaken, deteriorate or break (such as tree branches or vines), suitable artificial replacements may provide the necessary environment for the behaviour and be a considerably more durable, sturdy or clean provision, which would require far less maintenance (and therefore monetary cost). As is becoming apparent in novel affective state research, interactions with humans may actually be beneficial and rewarding for *some* zoo-housed species in *some* situations [38,39,40,41]. If a zoo is too focused on *wilding* their animals, opportunities to truly provide the best positive welfare conditions for the captive animals may be missed or ignored. Therefore, mixed natural/artificial enclosures for animals in zoos, that consider function, aesthetics, appropriate contact with humans, and practicality, may be much more fitting than the natural-only enclosures of the recent past. Two questions we might ask ourselves of mixed natural-artificial environments are as follows:Does the animal have the capacity to *know* that the environment is (partly) artificial?Does the animal *care* if the environment is (partly) artificial?

These are open-ended questions that might be addressed in a separate paper, drawing from current knowledge of animal neurobiology and cognition, and their needs and wants for a “life worth living” [4,42]. There is some evidence that some species do indeed display an innate preference for naturalistic “enriched” enclosures as opposed to basic artificial environments without many features (barren environments) (Box Turtles [43,44]; Coal tits and blue tits [45]), suggesting that some animals may indeed have a capacity to identify *natural* environments. Alternatively, perhaps they just innately prefer *non-barren*, enriched environments—perhaps these animals would be just as likely to select enriched *artificial* environments over any basic or barren environments. Utilising current animal welfare research and expert consensus a new era of evidence-based enclosure design, natural or not, which consider the animals’ needs foremost, should be the next step forward for zoo institutions [46]. As will be explored later, unnatural or artificial environments can still be compatible with promoting the expression of natural behaviours.

A dilemma with *wilding*, then, is that attempts to treat captive animals as wild are partly or wholly incongruent with their actual situation. As has been said in this article before, captive animals are not, nor will they be, *wild animals living in a wild environment*. Their living environment is completely curated by humans, who must make many decisions for the animals for their best interests. This does not mean that we should attempt to treat all captive animals as we would extensively domesticated animals such as livestock or companion animals (i.e., dogs and cats). Zoos indeed strive to keep their wild animals as “undomesticated” as possible [12]. This, however, may be an unattainable ideal, due to unintended and unavoidable genetic and epigenetic drifts favouring adaptations for life in a captive environment, despite our best efforts to otherwise prevent this from occurring. Indeed, in a human-animal interaction review chapter, Hemsworth et al. [18] write about the possibility of unintended domestication in zoos, citing research such as Price [47,48]—“While zoo animals are generally not considered to be domestic animals, domestication can obviously occur with wild animals kept and bred in captivity, such as zoos, but the extent of the domestication process will depend on the rate of artificial selection” [18]. The chapter also highlights the distinction between domestication of a group of animals, and *taming* of an individual animal—domestication can be defined as “a process by which a population of animals becomes adapted to man and to the captive environment by genetic changes occurring over generations and environmentally induced developmental events reoccurring during each generation” [48]; whereas taming is simply “an experiential (learning) phenomenon occurring during the lifetime of an individual animal” [47]. Domestication is a process most likely to happen to animals that are purposefully kept in captivity, and artificially bred or selected, or genetically altered, by humans. Individual taming may more frequently occur in both captive *and* wild animals that are in regular contact with humans.

To unpack this, we should consider other historical animal domestications. The domestication process has taken thousands of years for those animals that we now consider domesticated. In that time, these animals have been subject to multiple selective pressures including artificially imposed selective breeding, turning them from a “wild-variant” into domesticated animals, specifically chosen for their desirable adaptations. A strong argument against the concept of natural living for these domesticated animals, therefore, is that these animals don’t actually represent or reflect any animal which may be found in the *wild* or in *nature* [10]. They have transformed into animals that don’t fill any natural ecological niche, whose existence is solely reliant upon human intervention and care, and their persistence is reliant upon humans’ continual propagation of that lineage. Of course, if all human interference or interaction were to cease, these “unnatural” animals are still a part of the *biotic community* of Earth, and they would be able to freely breed and propagate themselves. Yet still they would not be a part of the current *natural ecosystem*, they still would not have a natural ecological niche, and many cases of free-living livestock or pets (feral animals) in many inappropriate locations have led to irreparable habitat degradation or even ecosystem collapse [28,30].

Many researchers posit the co-evolution of wolves and humans, rather than the one-way domestication of the animal [49]. Both species adapted to working with each other (for the benefit of both) over tens or hundreds of thousands of years. Wolf-human co-evolution is now suggested to have happened at multiple historical intervals in different geographical regions, leading to the rise of an entire species (or sub-species), dogs (*Canis familiaris*, or *Canis lupus familiaris*), and a multitude of breeds [49]. This co-evolution theory may plausibly explain the domestication process of most modern livestock and pets. Novel research also suggests that the co-evolution of humans and many of our domestic species may have been modulated and propagated by the shared experience of bonding, through the ubiquitous neurotransmitter *oxytocin* [50,51]. While general consensus would not consider zoo-housed animals as domesticated, we must consider that humans have unintentionally started these animals down a similar domestication pathway, as we now approach the third century of keeping animals in zoos, with many captive animal lineages able to be traced back over 100 years in captivity [12]. This generational captive breeding (including artificial selection of mates) will certainly have profound effects on the prevalent adaptations of these captive animals—adaptations to life in a captive environment and in close proximity with humans. Speculatively, it is possible that close contact with humans may be activating oxytocin pathways in many captive zoo species, leading to positive affiliative (or bonding) human-animal interactions. Indeed, some researchers are starting to focus on reported keeper-animal bonds in zoos [52,53]. However, 300 years is still a shorter timespan than the domestication process for most other animals we keep today (with exception for some farmed species, such as rapidly “domesticated” mink and foxes), and most animals displayed in zoos still resemble and behave like their wild counterparts far more than any newly bred type of domesticated animal.

One of the core tenets of zoos is to display *wild* animals that have, and will retain, a certain *wildness* to visitors, not to breed new types of domesticated animals [12]. Therefore, many practices and safeguards are employed by zoos to try to maintain this *wildness*. However, the efficacy of our attempts to retain wildness may eventually be mooted by uncontrollable selective pressures of generational life in captivity. If zoos exist 1000 years from now, zoo animals may have significantly drifted from true representations of their wild counterparts (many of which will be extinct in the wild). But, zoos will still strive to maintain wildness. And for many animals, zoos’ careful management will at least succeed in *slowing* the rate of domestication, but inevitably some genetic or epigenetic drift (mitochondrial drift), or even morphological drift, might still occur regardless of our procedures and safeguards. Thus, these captive animals that still resemble wild species must have specific requirements for care and housing that may differ from common practices for domesticated animals. This is the care that zoos should, and do, provide. But zoos must also make many ethical judgements and decisions which will benefit the animal for a full and rich life in captivity, whether wild or domesticated or somewhere in-between.

There is significant pressure on zoos to exist to advance both animal welfare and wildlife conservation priorities. Indeed, the World Association of Zoos and Aquariums (WAZA) cite conservation as zoos’ core *purpose*, but fostering positive animal welfare is their core *activity* [32]. However, this animal welfare strategy document also quite plainly acknowledges that often conservation priorities may compromise optimal welfare, but zoos should always endeavour to minimise welfare-reducing conditions [32]. A strong priority of zoos is to avoid genetic drift towards domestication of their captive held *wild* animals, but, as explained above, there is still a risk that time will change these animals in unknowable ways. This is not written intentionally as an inflammatory argument against genetic selection and diversity processes utilised by zoos, but merely as an acknowledgement of the inherent entropy of many natural systems, and an acknowledgement that humans do not have absolute control of natural processes. But, we do our best with the science and technology that we have available. This ethical *wildness* dilemma has been explored in context of other arguments, such as human-controlled *facilitated adaptation* to climate change impacts [29]. It should also be considered that there may be negative impacts of zoos maintaining *wildness* in their non-releasable captive animals, especially in species known to have low behavioural plasticity [30]. For example, some wild animals may be very prone to negative welfare states due to captivity, manifesting in fear or anxiety responses and behavioural patterns [24,30], whereas domesticated or semi-domesticated species (or wild species with high behavioural plasticity) may potentially cope better in captive environments [30].

## 5. (Un)Natural Behaviours

Part of the *natural living* concept is a focus on *allowing animals to express natural behaviours*. As has been pointed out by many, however, the definition of *natural behaviour* is problematic, especially when referring to domesticated species with no *natural* or *wild* equivalent animal, and therefore, no known *natural* behaviours (for review, see [10]). Again, *wilding* runs into problematic territory here, by over-emphasising or reinforcing only those *natural* behaviours that are generally displayed by the species in the *wild*. Academics have suggested multiple alternative terms for *natural behaviour* that may better define what is intended, such as *normal behaviours* or *species-typical behaviours* [1,10,18,34]. However, these terms still struggle to articulate which behaviours are definitely included as natural, and behaviours classified in this way may be adaptive or maladaptive for a captive environment. For example, migratory behaviour would be considered *normal* or *typical* for a migratory bird species, but is maladaptive in captivity as the animal can not fulfill that motivation [30]. Many behaviours that are displayed by a species in nature have no function or purpose in a captive setting. Simply because a natural behaviour is *not* displayed in captivity does not infer that the animal is in a state of distress or suffering. If a natural behaviour serves no purpose for the animal in its captive environment, the motivation to perform the behaviour may be very low or non-existent [19,23,24,34].

Therefore, more important measures of welfare-positive behaviours for captive animals are *highly-motivated behaviours*, and *highly-rewarding behaviours*. These behaviours may be part of a natural repertoire, or wholly *unnatural*—only displayed in captivity. So-called unnatural behaviours may be the most adaptive for the animal’s captive environment, and may be important for positive affective experiences for that animal. *Unnatural behaviours* do not fit with the ethos of the concept of *natural living* or *wilding*, and attempts may be made to extinguish these behaviours. However, this may actually be of more harm to the animal than benefit—if the behaviour is highly motivated, frustration of that motivation may lead to a negative affective state, and possibly a negative welfare state [24]. Restricting an animal’s behaviours to only those which are considered natural may also significantly reduce that animal’s ability to make choices (reducing self-determined agency), which in turn leads to a perceived lack of control over their situation, which is known to negatively affect coping efforts and welfare of captive animals [31,54,55,56].

Another curiosity of *nature* is what I will term *unexpected natural behaviours*. These are behaviours that will be performed by wild animals in specific unnatural situations, such as interacting with artificial running wheels or mirrors placed in wild environments. Quite a few “popular science” documentaries and online videos show the effects of placing these sorts of objects in nature. Often animals in these videos will run in the artificial wheel, or stare at their reflection for long periods [57]. These are wholly *wild* animals that are interacting *naturally* with artificial (*unnatural*) objects. Following from this, many behaviours in captive animals may be incorrectly classified as *unnatural*, as they are behaviours that are also displayed by wild animals with access to the same or similar unnatural objects.

To increase well-being and assist positive welfare outcomes for captive animals, focus needs to shift from a fixation on what are considered *natural behaviours* to those behaviours which the animal appears *highly-motivated* to perform. Thus, rather than focus on *treating animals as though they were wild*, it would be more pertinent to focus on *allowing animals to express highly-motivated behaviours*, particularly if deprivation or frustration of these behaviours results in significant stress, reduced fitness and/or a negative affective state [24]. Expressing highly-motivated behaviours may also afford the animals more agency and choice within their environments [54], which should be allowed within reasonable limits—the allowed behaviours must not compromise the safety or health of the individual performing the behaviours, or of the other animal(s) or human(s) involved (i.e., allowing a predator to hunt for live prey does not consider the ethical obligations for the safety of the intended prey animal). This may be categorised into *harmless* and *harmful wants* of an animal. *Harmless wants* may include highly-motivated behaviours such as foraging, climbing, playing or resting. *Harmful wants* may include highly-motivated behaviours such as feeding, hunting or fighting *without restriction*. The important factor here is that *harmful wants without restriction* can lead to *harmful consequences* (negative, self-injurious or self-destructive outcomes) for the individual performing the behaviour, or for individuals that are the target of the behaviours. A classic example is allowing Labradors access to food *ad libitum* will often result in excessive overeating causing multiple long-term health problems, such as obesity and other related conditions. These limits need to be examined carefully and thoroughly, as they will be very species- and individual-specific behavioural limitations. Many zoos are already doing this, however consensus for an ethical and practical realignment towards promoting *highly-motivated behaviours* instead of *natural behaviours* needs to be agreed to and endorsed by zoological institutions, associations, workers and allies.

## 6. Are Human-Animal Interactions Natural?

Whether zoos focus on natural behaviours or highly-motivated behaviours, both of these may still include direct interactions with humans. It is often supposed or assumed that human-animal interactions in zoos are an unnatural phenomenon, however, there is one clear way to counter this presumption. In nature, wild animals encounter many other species around them, to which they must adapt, and often interact with, in positive, negative and neutral ways (from symbiotic relationships to parasitic or predatory relationships). Most wild animals now have to adapt not only to their historically natural ecosystem conspecifics, but also to a multitude of invasive species that were previously unknown to them or their ancestors [28,30]. Also, as we now live in the *Anthropocene* era, wild animals increasingly have to adapt to the ever-growing and ever-encroaching human population, in an increasingly human-affected world [28,30,33]. In captivity, then, are not humans one of those species to adapt to, and to interact with? Humanity often assumes some removal of our species from the rest of nature, that we are somehow a step apart from other animals. It is doubtful that this is how other animals view humans, however. Often one of the great curiosities of the natural world is how competing animal species may form symbiotic balances that benefit all, and actively help each other in interactions. These would be deemed *natural behaviours*. Therefore, if many species actively interact with other species as a mode of adaptation to their environment, would it not follow that human-animal interactions in zoos could actually be considered quite *natural* adaptations? And if those interactions are highly-motivated in the animal, should we encourage them?

Whether these interactions are deemed *natural* or *unnatural*, allowing for positive human-animal interactions may be one avenue of increasing positive affective experiences for animals, especially if those animals are highly motivated to interact with humans (whether it be zookeepers or zoo visitors) [39]. These interactions must be subject to rigorous safety evaluations for all participants, of course. However, the current status quo of *wilding* frameworks often view these interactions as undesirable in any and all situations, regardless of the animal’s motivations behind the intended behaviours. Again, frustration of these motivations may actually be detracting from an animal’s well-being. If an animal is *highly motivated* to interact with humans in or around its environment, and if those interactions are considered *safe* for all participants, then those interactions should be allowed to occur, or even promoted (through *supervised* offerings of such interactions). Obviously some interactions are exempt from these stipulations, when considering an animal’s overall health or best interests (such as veterinary procedures or restraint for medical treatment), though positive reinforcement training schedules can often remove some of the harshest penalties to the animals these situation might present (such as training for quick, mildly-aversive hand-injections, blood sampling, or “crate training” for restraint and transport) [37,58].

An animal’s motivation to engage in positive human-interactions may vary from day-to-day, based on other internal and external factors, but the animal should never be confined to, or negatively coerced into, an interaction scenario. The choice to interact should always be on an animal’s own terms. This may not be the case for all human-animal interactions currently deployed by zoos across the world. Often, many “encounter” or “interaction” animals are not afforded a choice of whether to participate or not, or are housed in inadequate areas that may increase their desire to escape that area, even if it means having to interact when they are unwilling [59,60]. Most industry-accredited zoos have their own welfare charter, and have processes and policies implemented to safeguard encounter animal well-being, and to try to offer as much choice as possible to the animals before being handled for interactions. Indeed, the guidelines published by WAZA [32] state that: “*Interactive experiences should be non-invasive, safe and non-stressful for animals. Monitoring of all animals involved in interactions must be ongoing and have professional oversight. Risks to animal welfare should be minimised by carefully considering whether interactive experiences are appropriate, and if they are, by accommodating the animals’ particular needs*” (p. 74).

## 7. Conclusions

*Natural living* may be a useful concept for developing robust measures of holistic zoo animal welfare, but care must be taken to avoid the pitfalls and dilemmas explored in this article. Specifically, *wilding* is a concept that may not truly be providing zoo personnel with an appropriate ethical or conceptual basis for optimizing evidence-based animal welfare. Zoos will continue existing well into the future, and so more appropriate measures of *what is important to an animal* for a “life worth living” in captivity should tend towards *highly-motivated behaviours* rather than just *natural behaviours*. *Human-animal interactions* in zoos are a source of debate and controversy, however, if implemented appropriately, they may significantly enhance animal well-being and holistic animal welfare (which may still be distinctly different concepts, even though the words are now often used interchangeably [10]), as they are often relationships of great importance to captive animals. Further exploration of what might constitute positive human-animal interactions, both scientifically and ethically, as well as ways of implementing such interactions without leading to unintended or “undesirable” human behavioural patterns emerging (such as an increased desire to “own” exotic wildlife) shall be forthcoming as a follow-up to this article.

## Figures and Tables

**Figure 1 animals-09-00318-f001:**
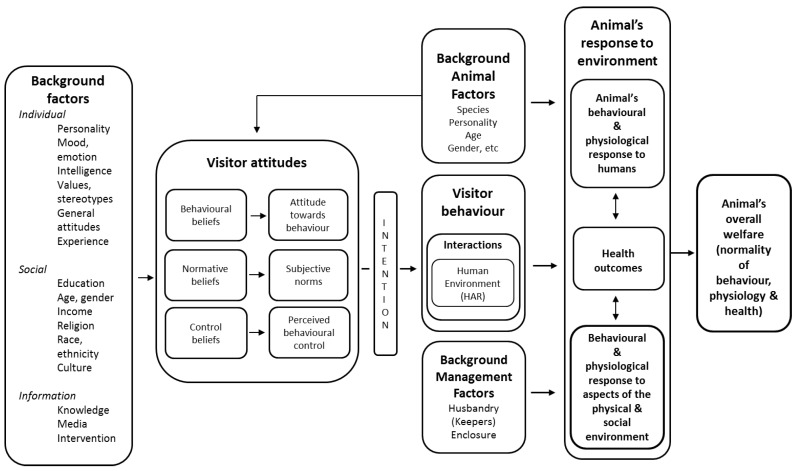
Proposed visitor-animal interaction model (adapted from Hemsworth-Coleman model (2011) by S. Chiew and L. Hemsworth, *pers. comms.*, 2016) [20].

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
