# Peer review of "Dilemmas for Natural Living Concepts of Zoo Animal Welfare"

_animals, 2019, doi:10.3390/ani9060318_

Round 1

Reviewer 1 Report

This is an interesting discussion of the ethical discourse associated with zoo animal welfare conceptualisations. The main problem I have with this manuscript is the lack of references to back-up key points you have made. These are not original arguments. I have marked areas in the attached document where references must be included.

These are also some suggestions I have:

Please reconsider stating that it is 'agreed by many' that zoos are not ethically wrong in the simple summary and abstract - this is clearly your opinion (as evidenced in L61).

Please correctly refer to the three 'orientations' to animal welfare on L46 and 50. Also, reference to Fraser 1997 should be included.

The Five Domains Model should be capitalised and it does not 'define' animal welfare, but does assist in 'characterising' it.

Please clarify what the 'empathetic or kindness-based approach' to welfare is and define/clarify compassionate conservation.

L80-83 makes it sound like relationships are not reflected by affective states, however, feelings of congeniality/affiliative bonding can result in positive affective engagement - this has been well referenced in the literature.

Please provided an example at L103. New paragraph at L104.

Section 3 is woefully lacking in references. I have identified a large number of places where references need to be included for statements made. See attached document.

L141-142, you are conflating ethics and welfare. Stick to discussing welfare in this paragraph.

L175, please provide an example of routine husbandry procedures in zoos to illustrate your point.

L319, please add something that indicates the negative impact of zoos maintaining 'wildness' in their captive populations i.e. the resultant fear/anxiety/boredom - versus domesticated zoo animals that could be better adapted to zoo life

L324, domesticated species may still have natural behaviours, we are just not necessarily aware of them. Please change 'actual' to: 'known'

L329, please provide an example

L370, ad libitum should be italicised

L370-371, where is your evidence for Labrador's stomach bursting? The reference typically used for this refers to Labradors overeating resulting in obesity and related disease (poor biological function at the expense of natural living and behavioural agency).

Author Response

I thank the reviewer for taking the time to review my manuscript, and thank them for the feedback. I have addressed all comments where possible, and added extra references for some sections. The reviewers comment were very helpful for improving my manuscript. Please see below a summary of my changes and responses as indicated by red, bold text under each of the reviewer's comments. I have also uploaded a “tracked changes” copy of the manuscript in PDF form.

Reviewer 1 comments

Please reconsider stating that it is 'agreed by many' that zoos are not ethically wrong in the simple summary and abstract - this is clearly your opinion (as evidenced in L61).

L15 Changed to ‘some’, as this is a reference to other ethical works by Palmer & Sandøe and Palmer, Morrin & Sandøe, not just my opinion.

Please correctly refer to the three 'orientations' to animal welfare on L46 and 50. Also, reference to Fraser 1997 should be included.

L63-64 Changed to “three conceptual frameworks (orientations)” as taken from reference 2- Hemsworth, et al. 2015. Reference to Fraser 1997 has been added here as well.

The Five Domains Model should be capitalised and it does not 'define' animal welfare, but does assist in 'characterising' it.

L69 Agreed. Changes made.

Please clarify what the 'empathetic or kindness-based approach' to welfare is and define/clarify compassionate conservation.

L58 First part deleted for brevity. Compassionate conservation referenced as work of philosopher Bekoff 2013, as applied and adapted by Gray 2017.

L59-62 Added sentence: “At the moment compassionate conservation remains very anti-zoo in its position, however, as Gray (ref) posits, there is much merit in using this ethic to work with zoos constructively, to enhance zoos' ethics and practices.”

L80-83 makes it sound like relationships are not reflected by affective states, however, feelings of congeniality/affiliative bonding can result in positive affective engagement - this has been well referenced in the literature.

L82-L88: Various wording changes made to include this suggestion, including new sentence: “These relationships may then be reflected by the internal affective states of both (or all) agents in that interaction.”

Please provided an example at L103. New paragraph at L104.

New paragraph change made. For brevity, a zoo example was not included.

Section 3 is woefully lacking in references. I have identified a large number of places where references need to be included for statements made. See attached document.

References cannot be provided in a few places – this is my original thought and logical statements. References have been added where able/appropriate.

L125-126 Reference added for definition of “wild”.

L127 Sentences added: “From a decade of first-hand experience within the zoo industry, this wilding conception of natural living has been encountered often enough to be considered pervasive amongst many zoo personnel's implicit beliefs and taught knowledge about how zoos should approach animal welfare, though actual prevalence rates have not been systematically investigated. Indeed, many welfare assessment and monitoring tools deployed by zoos focus somewhat on natural environments and natural behaviours (Sherwen et al., 2018)”.

L135 “suppose as” changed to “presume”.

L141-142, you are conflating ethics and welfare. Stick to discussing welfare in this paragraph.

This paragraph is about ethics and moral “rightness” of our actions, using welfare-compromising hardships of nature as examples. It is not meant to conflate ethics with welfare.

L151-152 added “(morally)” in sentence “Would it be right?”

L153-154 changed words in parentheses to “(thereby destroying a natural habitat and causing displacement of many native species)”

It was not considered necessary to add references for common facts of the history of Earth, such as the rise of life and ecological changes.

L159 references added.

L168 references added.

L174 references added.

L178-L180 sentence added: “Indeed, if captivity is providing all of the needs and wants of an animal (including positive affective experiences), but without liberty, then liberty is not necessarily a basic interest of the animal [4]”.

L182-L185 sentence added: "This is an ever more salient point after the United Nations Intergovernmental Science-Policy Platform on Biodiversity and Ecosystem Services (IPBES) released a 2019 report which estimates that anthropogenic influences may cause the extinction of 1 million species of animals and plants (IPBES, 2019)".

L175, please provide an example of routine husbandry procedures in zoos to illustrate your point.

L192-L194 Sentence added: “For example, it has been reported that reliably signalling startling husbandry events can improve stress resilience and welfare of zoo-housed capuchins (Sapajus paella), whilst still leading to physiological arousal within the animals (Rimpley, 2013)”

L211 references added.

L213 added words “arousal and” to sentence: “…to ensure that the intended stress resilience…”

L222 reference added.

L289 references added.

L291 reference added.

L302 reference added.

L312 reference added.

L319, please add something that indicates the negative impact of zoos maintaining 'wildness' in their captive populations i.e. the resultant fear/anxiety/boredom - versus domesticated zoo animals that could be better adapted to zoo life

L338-343 Sentences added: “It should also be considered that there may be negative impacts of zoos maintaining wildness in their non-releasable captive animals, especially in species known to have low behavioural plasticity (Mason 2013). For example, some wild animals may be very prone to negative welfare states due to captivity, manifesting in fear or anxiety responses and behavioural patterns (Mason 2010, Mason 2013), whereas domesticated or semi-domesticated species (or wild species with high behavioural plasticity) may potentially cope better with captive environments (Mason 2013).”

L324, domesticated species may still have natural behaviours, we are just not necessarily aware of them. Please change 'actual' to: 'known'

L348 Change made.

L329, please provide an example

L354-355 example added.

L376 reference added.

L370, ad libitum should be italicised

L397 Change made.

L370-371, where is your evidence for Labrador's stomach bursting? The reference typically used for this refers to Labradors overeating resulting in obesity and related disease (poor biological function at the expense of natural living and behavioural agency).

Agreed. This was an errant sentence incorrectly added from an earlier draft.

L396-398 Change to sentence made: “allowing Labradors access to food ad libitum will often result in excessive overeating causing multiple long-term health problems, such as obesity and other related health conditions.”

L383 reference added.

L411 & L413 references added.

L436 references added.

Reviewer 2 Report

This is a well-written manuscript exploring the concepts of natural living and wilding in relation to the welfare of captive animals. That author makes a convincing argument as to how these concepts may not be the best basis for promoting animal welfare. I would however, consider some re-organization to help with the flow of the article for readers. For example, switching the second and third paragraphs in the introduction would allow for for better transitions. I would also consider switching sections 3 and 4 ("Wilding: the Natural Living Dilemma and "Natural living or just Natural Looking). The concept of wilding, as stated by the author, does not fit into the zoological model. It would be helpful if how/where this concept is being applied or promoted could be included here. The author cautions readers against using wilding as a means to manage animals in captivity and as a way to assess the welfare of those animals. However, it is unclear if this is being applied by some in the global zoological community, and if it truly a "risk". Several very important points are made in this article, including the fact that displaying natural behaviors does not equate to positive welfare, that a more appropriate standard may be to use the term highly-motivated behaviors and that some negative experiences can actually enhance overall welfare. The author should, however, specifically address the fact that some highly-motivated behaviors may be harmful to the individual animal (i.e., stereotypies or self-injurious). This is somewhat superficially addressed, but concentrates on behaviors that could be harmful to others (e.g., hunting). It would also be worth noting that human-animal interactions, even if animals are motivated to participate in them, may lead to zoo visitors having increased expectations or desire to obtain such animals as companions.

Author Response

I thank the reviewer for taking the time to review my manuscript, and thank them for the feedback. I have addressed all comments where possible, and added extra references for some sections. The reviewers comment were very helpful for improving my manuscript. Please see below a summary of my changes and responses as indicated by red, bold text under each of the reviewer's comments. I have also uploaded a “tracked changes” copy of the manuscript in PDF form.

Reviewer 2 comments

This is a well-written manuscript exploring the concepts of natural living and wilding in relation to the welfare of captive animals. That author makes a convincing argument as to how these concepts may not be the best basis for promoting animal welfare.

 I would however, consider some re-organization to help with the flow of the article for readers. For example, switching the second and third paragraphs in the introduction would allow for better transitions. I would also consider switching sections 3 and 4 ("Wilding: the Natural Living Dilemma and "Natural living or just Natural Looking).

Agreed. Second and third introduction paragraphs switched. Sections 3 and 4 have not been switched.

The concept of wilding, as stated by the author, does not fit into the zoological model. It would be helpful if how/where this concept is being applied or promoted could be included here. The author cautions readers against using wilding as a means to manage animals in captivity and as a way to assess the welfare of those animals. However, it is unclear if this is being applied by some in the global zoological community, and if it truly a "risk".

L127-L131: Added sentences: “From a decade of first-hand experience within the zoo industry, this wilding conception of natural living has been encountered often enough to be considered pervasive amongst many zoo personnel's implicit beliefs and taught knowledge about how zoos should approach animal welfare, though actual prevalence rates have not been systematically investigated. Indeed, many welfare assessment and monitoring tools deployed by zoos focus somewhat on natural environments and natural behaviours (Sherwen et al., 2018)”.

Several very important points are made in this article, including the fact that displaying natural behaviors does not equate to positive welfare, that a more appropriate standard may be to use the term highly-motivated behaviors and that some negative experiences can actually enhance overall welfare. The author should, however, specifically address the fact that some highly-motivated behaviors may be harmful to the individual animal (i.e., stereotypies or self-injurious). This is somewhat superficially addressed, but concentrates on behaviors that could be harmful to others (e.g., hunting).

L394-398 Sentences changed to incorporate this point: “The important factor here is that harmful wants without restriction can lead to harmful consequences (negative, self-injurious or self-destructive outcomes) for the individual performing the behaviour, or for individuals that are the target of the behaviours. A classic example is allowing Labradors access to food ad libitum will often result in excessive overeating causing multiple long-term health problems, such as obesity and other related conditions.”

It would also be worth noting that human-animal interactions, even if animals are motivated to participate in them, may lead to zoo visitors having increased expectations or desire to obtain such animals as companions.

Agreed. This is to be discussed in the forthcoming follow-up article.

L461 Sentence changed: “Further exploration of what might constitute positive human-animal interactions, both scientifically and ethically, as well as ways of implementing such interactions without leading to unintended or "undesirable" human behavioural patterns emerging (such as an increased desire to "own" exotic wildlife) shall be forthcoming as a follow-up to this article.”

Round 2

Reviewer 1 Report

The author has made significant changes to the manuscript and it is now a very interesting discussion of this topic.